# Effects of Thiamethoxam and Fenvalerate Residue Levels on Light-Stable Isotopes of Leafy Vegetables

**DOI:** 10.3390/foods12142655

**Published:** 2023-07-10

**Authors:** Fang Qi, Xing Liu, Zhongsheng Deng, Yangyang Lu, Yijiao Chen, Hao Geng, Qicai Zhang, Qinxiong Rao, Weiguo Song

**Affiliations:** 1Institute for Agro-Food Standards and Testing Technology, Shanghai Academy of Agricultural Sciences, Shanghai 201403, China; qifang3028@163.com (F.Q.); liuxinglyg@126.com (X.L.); zhongshengdeng668@163.com (Z.D.); yyanglz@163.com (Y.L.); cx1213097626@163.com (Y.C.); genghao0128@163.com (H.G.); qicaizhang@126.com (Q.Z.); qinxiongrao@163.com (Q.R.); 2College of Food Sciences, Shanghai Ocean University, Shanghai 201306, China; 3Shanghai Service Platform of Agro-Products Quality and Safety Evaluation Technology, Shanghai 201403, China

**Keywords:** thiamethoxam, fenvalerate, spinach, cabbage, lettuce, light-stable isotopes, relationship, PLS-DA

## Abstract

Accurate identification of the rational and standardized use of pesticides is important for the sustainable development of agriculture while maintaining a high quality. The insecticides thiamethoxam and fenvalerate and the vegetables spinach, cabbage, and lettuce were used here as study objects. Descriptive analysis and primary reaction kinetic equations were used to analyze the changes in metabolic residues of the two insecticides after different numbers of application in three vegetables. The effects of pesticide residue levels on the *δ*^13^C, *δ*^15^N, *δ*^2^H, and *δ*^18^O values of vegetables were analyzed by one-way analysis of variance and correlation analysis. Partial least squares discriminant analysis (PLS-DA) was applied to build discrimination models of the vegetables with different pesticide residues based on stable isotopes. The results showed that the first degradation residues of thiamethoxam and fenvalerate in spinach, cabbage, and lettuce conformed to primary reaction kinetic equations, but the degradation half-lives were long, and accumulation occurred in the second application. The differences in the four stable isotope ratios in the control group of the three vegetables were statistically significant, and two-thirds of the stable isotope ratios in the three vegetables with different numbers of pesticide applications were significantly different. The *δ*^13^C and *δ*^15^N values of spinach, the *δ*^13^C, *δ*^15^N, and *δ*^2^H values of cabbage, and the *δ*^13^C, *δ*^15^N, *δ*^2^H, and δ^18^O values of lettuce were significantly correlated with different residues of thiamethoxam and/or fenvalerate applications. The control groups of the three vegetables, spinach-thiamethoxam-first, spinach-thiamethoxam-second, cabbage-thiamethoxam-second, cabbage-fenvalerate-first, and lettuce-thiamethoxam-first, were fully identified by PLS-DA models, while the identification models of other vegetables containing pesticide residues still need to be further improved. The results provide technical support for identifying the rational use of pesticides in vegetables and provide a reference method for guaranteeing the authenticity of green and organic vegetables.

## 1. Introduction

The rational use of chemical pesticides can ensure agricultural product output and improve agricultural economic benefits, but frequent and improper use not only leads to crops and pests developing resistance but also excessive residues in crops and environmental pollution [1]. In particular, some chemical pesticides with long degradation cycles that are semi-volatile, toxic, and high bioaccumulation can pollute the atmosphere, water, and soil, and they gradually accumulate in organisms through the enrichment effect of the food chain, further conferring adverse effects on the ecosystem and human health. With the development of agriculture through modernization, ecologically sustainable agriculture has become a key path. It is not only necessary to regulate the use of chemical pesticides, but also to select high-efficiency, low-toxicity and low-residue chemical pesticides, and even to replace chemical pesticides with biological pesticides or physical methods [2]. Green and organic agricultural products are the products of ecological sustainable development of agriculture and have higher prices than those of conventional agricultural products due to their safety and high quality, which also leads to frequent food fraud [3]. Furthermore, green product is a popular alternative to organic product in China, as defined by Chinese National Standard GB/T 33761-2017 and Agriculture Industry Standard NY/T 391-2013, and organic products were defined by Chinese National Standard GB/T 19630-2019. They are certified by specialized agencies and then given corresponding information labels. In addition to information traceability technology, chromatography and mass spectroscopy technology based on pesticide residue determination have also been applied to ensure their authenticity [4,5]. However, the information labels are easily disturbed by human factors [6], and the pesticide residue determination cannot effectively detect prohibited pesticides in a safe time interval [7], which causes difficulties in the authenticity identification of green and organic agricultural products.

The natural abundance of stable isotopes in agricultural products can objectively reflect information about the geographic origin, climatic environment and use of inputs during the planting process. For example, carbon isotope can reflect information about water stress, light intensity, and environmental pollution (such as automobile exhaust) during the growth process [8,9]; nitrogen isotope can reflect information about soil inputs (mainly fertilizers), air pollution (NH_3_ or NO_x_), and climatic conditions (such as rainfall and temperature) [8,10]; hydrogen and oxygen isotopes can reflect information about irrigation water and atmospheric water vapor [8,11]. Stable isotope technology has been widely used in geographic origin traceability and cultivation method discrimination of agricultural products and has become an effective means to trace the source of environmental pollutants [12,13,14]. For example, compound-specific stable isotopes have been used for tracing of polycyclic aromatic hydrocarbons, methyl tert-butyl ether, and other environmental pollutants [15,16].

Pesticides, as a common agricultural input, may affect the natural abundance of stable isotopes in agricultural products [17]. Therefore, it is feasible to use the change in stable isotopes to determine the application of pesticides in agricultural products, which can provide a technical reference for the identification of green and organic agricultural products. However, to the best of our knowledge, there are no research reports on the relationship between pesticide residues and stable isotope ratios in agricultural products. This study aims to investigate the relationship between different insecticide residues and light-stable isotope ratios (*δ*^13^C, *δ*^15^N, *δ*^2^H, and *δ*^18^O) by continuous application of thiamethoxam (a widely used neonicotinoid pesticide) and fenvalerate (a lipid-soluble pyrethroid pesticide) on the common leaf vegetables cabbage, lettuce, and spinach.

## 2. Materials and Methods

### 2.1. Chemicals and Reagents

A thiamethoxam standard (purity 99.6%) and fenvalerate standard (purity 99.7%) were purchased from Dr. Ehrenstorfer GmbH (Augsburg, Germany), and the structural formulas of the two pesticides were shown in Figure 1; 25% thiamethoxam water dispersible granules were supplied by Syngenta (Shanghai, China) Investment Co., Ltd.; and a 20% fenvalerate emulsion was purchased from Jiangsu Huifeng Bio-Agriculture Co., Ltd. (Shanghai, China). A compound chemical fertilizer was supplied by Hubei Yihua Group Co., Ltd. (Yichang, China). Acetonitrile, methanol, formic acid, and ammonium acetate were of HPLC-grade and purchased from Merck KGaA (Darmstadt, Germany). Sodium chloride, hexane, and acetone were of analytical grade and purchased from Sinopharm Chemical Reagent Co., Ltd. (Shanghai, China). Florisil solid phase extraction column, amino solid phase extraction column and 0.22 μm organic filter membrane were purchased from ANPEL Laboratory Technologies Inc. (Shanghai, China). Stable isotope reference material IAEA-603 was provided by the International Atomic Energy Agency (Austria), and USGS40, USGS54, USGS90, and USGS91 were provided by the United States Geological Survey (Sunrise Valley Drive Reston, VA, USA).

### 2.2. Field Experiment

Cabbage, spinach and lettuce were planted at the Zhuanghang Comprehensive Test Station of Shanghai Academy of Agricultural Sciences from November 2019 to January 2020. The same organic manure was applied to the three vegetables as the base fertilizer, and the planting area was 10 m^2^. The cabbage was transplanted to a defined location, and thiamethoxam and fenvalerate were applied after 60 days, and a compound chemical fertilizer was applied once between transplanting and before pesticide applications. The pesticides were applied to the lettuce 30 days after transplanting, and were applied the spinach 30 days after sowing. The pesticides were applied twice with an interval of 5 days. The first application of thiamethoxam and fenvalerate were evenly sprayed at 0.03 g/m^2^ and 0.07 mL/m^2^, respectively. The dosage of the second application was doubled so that thiamethoxam and fenvalerate were evenly sprayed at 0.06 g/m^2^ and 0.14 mL/m^2^, respectively. At 2 h, 24 h, 72 h, and 120 h after application, triplicate samples of each vegetable were collected, and each sample was 1 kg.

### 2.3. Sample Preparation

The collected vegetable samples at the time set for the experiment were cleaned and pulped. Some of vegetable samples were frozen at −18 °C for 6 h and freeze-dried at −54 °C for at least 72 h. The dried vegetable samples were finely ground to a homogeneous powder (<0.15 mm) before analysis. Some of the pulped vegetable samples were used for pesticide residue contents determination.

Thiamethoxam and fenvalerate in cabbage, spinach and lettuce were extracted and purified according to Chinese National Standard GB/T 20769-2008 and Chinese Agri-culture Industry Standard NY/T 761-2008.

### 2.4. Stable Isotope Analysis

Four stable isotope ratios (*δ*^13^C, *δ*^15^N, *δ*^2^H, and *δ*^18^O) of the vegetables, fertilizers, and pesticides were analyzed using a Flash IRMS elemental analyzer interfaced to a DELTA V Advantage isotope ratio mass spectrometry system (EA-IRMS, Thermo Fisher Scientific Inc., Dreieich, Germany) using similar methods to those we used previously [18]. Samples of about 0.8 mg and 3.5 mg were weighed tin capsules for ^13^C and ^15^N analysis, respectively, and 0.5 mg samples were weighed in silver capsules for ^2^H and ^18^O analysis, respectively. In the C/N mode, EA oxidation and reduction furnace temperatures were set at 980 °C. Helium was used as the carrier gas and with a flow rate of 180 mL/min, and carbon dioxide and nitrogen were used as the reference gas with a flow rate of 60 mL/min. In the H/O mode, EA pyrolysis was undertaken at 1380 °C. Helium was used as the carrier gas with a flow rate of 100 mL/min, and carbon monoxide and hydrogen were used as the reference gas with a flow rate of 100 mL/min. Isotope ratios were determined using Equation (1):(1)$$\delta X‰=\left(\frac{R_{sample}}{R_{standard}}-1\right)\times 1000$$
where X represents ^13^C, ^15^N, ^2^H, or ^18^O; *R*_sample_ is the abundance ratio of heavy isotope against light isotope, i.e., ^13^C/^12^C, ^15^N/^14^N, ^18^O/^16^O, and ^2^H/^1^H; and *R*_standard_ is the reference standard isotope ratio. Reference materials USGS40, USGS90, and USGS91 were used for *δ*^13^C and *δ*^15^N; IAEA-603, USGS90, and USGS91 for *δ*^18^O; USGS54, USGS90, and USGS91 for *δ*^2^H. A vegetable sample, selected as a quality assurance check and working standard, and blanks were included in each run. Instrument precision was lower than 0.1‰ for *δ*^13^C, 0.2‰ for *δ*^15^N, 2.0‰ for *δ*^2^H, and 0.5‰ for *δ*^18^O.

### 2.5. Thiamethoxam and Fenvalerate Residue Determination

The thiamethoxam contents in cabbage, spinach and lettuce were determined using an ACQUITY Ultra Performance liquid chromatography (Waters Corporation, Wood Dale, IL, USA) interfaced to a TSQ Quantum Ultra mass spectrometer (Thermo Fisher Scientific Inc., Waltham, MA, USA) according to Chinese National Standard GB/T 20769-2008. The fenvalerate contents were determined using an Agilent 6890N gas chromatograph (Agilent Technology Co., Ltd., Santa Clara, CA, USA) according to Chinese Agriculture Industry Standard NY/T 761-2008.

### 2.6. Statistical Analysis and Chemometrics Methods

Dynamic degradation curves and first-order kinetic equations of thiamethoxam and fenvalerate in vegetables were plotted and fitted by Matlab R2009a software (The MathWorks, Natick, MA, USA). Stable isotopes differences among the three vegetables and different applications of pesticides were indicated by boxplots, which were produced in Microsoft Office 365 Excel (Microsoft Corporation, Redmond, WA, USA). One-way analysis of variance (ANOVA) was applied to assess differences among the *δ*^13^C, *δ*^15^N, *δ*^2^H, and *δ*^18^O values of the three vegetables and between different applications (one or two) of the two pesticides in the three vegetables. A Pearson correlation coefficient was used to represent the relationships between stable isotope ratios and pesticide residue levels in the vegetables, and partial least squares discriminant analysis (PLS-DA) was used to identify vegetables with different concentrations of pesticides. ANOVA, Pearson correlation coefficient, and PLS-DA methods were also performed by using Matlab R2009a software (The MathWorks, Natick, MA, USA).

## 3. Results and Discussion

### 3.1. Characteristics of Pesticide Residues in Vegetables

Dynamic degradation curves of thiamethoxam and fenvalerate in lettuce, spinach, and cabbage were shown in Figure 2. After the first application, the pesticide residual amount in vegetables was the highest at 2 h. Lettuce had the highest content of thiamethoxam (6.63 mg/kg) and fenvalerate (8.36 mg/kg), followed by spinach and cabbage, which was attributed to lettuce leaves having a relatively larger specific surface area than spinach, and the wrapping structure of cabbage caused pesticide residues to accumulate on the outer leaves. The residue levels of fenvalerate in the three vegetables were higher than those of thiamethoxam. This might be attributed to the higher usage of fenvalerate (0.07 mL/m^2^) than thiamethoxam (0.03 g/m^2^) or to the structures of thiamethoxam and fenvalerate (Figure 1). For example, thiamethoxam has a negatively charged nitroimine pharmacophore group and a 2-chloro-5-thiazolyl group, which gives it good internal absorption conductivity and allows it to be easily metabolized in vegetables [19,20], while fenvalerate is a lipid-soluble pesticide, and its permeability might be relatively slow due to the effects of the waxy cuticle of vegetables [21,22]. The residual amounts of thiamethoxam and fenvalerate in the three vegetables showed a decreasing trend with extended time, and the degradation rates were slow. At 120 h after application, the degradation losses of thiamethoxam and fenvalerate were 49.11% and 27.57% in spinach, were 64.95% and 31.30% in cabbage and 44.90% and 40.84% in lettuce, respectively, which were lower than those of previous research [23,24,25,26]. For example, Zhao et al. [27] found that the degradation loss of thiamethoxam in spinach was more than 50% at 120 h after application, and Chen et al. [28] found that degradation loss of fenvalerate in spinach was more than 50% at 72 h after application. This could be mainly ascribed to the fact that thiamethoxam and fenvalerate were applied at the maturity stage of the three vegetables when vegetable metabolism had slowed, which caused slower pesticide degradation rates. In addition, the degradation loss rates of thiamethoxam were higher than those of fenvalerate in the corresponding three vegetables, which might be related to the different dosage and structure of pesticides [19,20,21,22].

The residue degradation of thiamethoxam and fenvalerate after the first application in spinach, cabbage and lettuce conformed to a first-order kinetic equation (Table 1). The correlation coefficients between the real values and the predicted values of the kinetic equations were statistically significant (*p* < 0.05) or extremely significant (*p* < 0.01), indicating that first-order kinetics were consistent with the degradation process of thiamethoxam and fenvalerate. All of the degradation half-lives (>73 d) of the two pesticides in the three vegetables were much longer than reported in the existing literature (1 d–7 d) [21,26,27,28], further conforming that the pesticide degradation ability of the three vegetables was weak due to the three vegetables being in a mature stage with slow metabolism, suggesting that pesticides application should be avoided during the mature stage of vegetables. The degradation half-life of the same pesticide in the three vegetables was different. The degradation half-life of thiamethoxam was consistent with the residual concentration in vegetables, with lettuce > spinach > cabbage, while the degradation half-life of fenvalerate was inconsistent with the pesticide residue levels. Lettuce, with the most residue, had the shortest degradation half-life, followed by cabbage with the least residue, and spinach had the longest degradation half-life. This could be due to thiamethoxam having good internal absorption conductivity. Its degradation metabolism might be mainly inside the vegetables, and therefore the degradation time was consistent with the pesticide residue levels [19,20]. Fenvalerate might mainly attach to the surface of vegetable leaves and therefore was degraded and metabolized in the environment, and a small amount might enter the interior of vegetables through the waxy cuticle of vegetables where it was degraded [21,22]. Furthermore, the thickness of the waxy cuticle of the three vegetables differed, which caused the degradation half-life of fenvalerate in the three vegetables to be inconsistent with the pesticide residue levels.

After the second application (Figure 2b), only the thiamethoxam residue level in lettuce decreased gradually over time, with the highest concentration (10.28 mg/kg) at 2 h. The highest residue levels of thiamethoxam and fenvalerate in spinach were at 24 h (11.21 mg/kg) and 72 h (11.74 mg/kg), and in cabbage at 72 h (0.55 mg/kg and 1.16 mg/kg), respectively. The highest residue level of fenvalerate in lettuce was at 24 h (22.98 mg/kg). These results indicated that thiamethoxam and fenvalerate accumulated in vegetables, possibly due to pesticide deposition in the closed environment of greenhouses. Because some pesticide droplets were suspended in the greenhouse space during application, they could not quickly exchange with the outside atmosphere due to the closed condition of the greenhouse, and instead slowly settled on the surface of the vegetable leaves [26], which caused the pesticide residue levels of some vegetables to increase with the extended time.

The residue levels of thiamethoxam and fenvalerate in the three vegetables after the second application were higher than those after the first application. At 2 h, the residue levels of thiamethoxam and fenvalerate in spinach and the residue levels of fenvalerate in lettuce were more than twice those in the first application due to the concentration of thiamethoxam and fenvalerate being doubled in the second application. Except for the similar residue levels of thiamethoxam and fenvalerate in spinach, fenvalerate residue levels were still higher than those of thiamethoxam, and the residue levels of thiamethoxam and fenvalerate in cabbage were still the lowest among the three vegetables (Figure 2), indicating that the residues of thiamethoxam and fenvalerate in the three vegetables were not only affected by the degradation and metabolic capacity of the vegetables and of the concentration, structure and properties of the pesticides, but also by the growing environment of the vegetables.

### 3.2. Characteristics of Stable Isotope Distribution in Vegetables

There were differences among the *δ*^13^C, *δ*^15^N, *δ*^2^H, and *δ*^18^O values of the three vegetable control groups (Figure 3). Lettuce had the highest δ^13^C value (mean −28.3 ± 0.3‰) and was significantly different from that of spinach and cabbage (*p* < 0.05), which could be attributed to the differences in light use efficiency among the three vegetables [7,8]. Compared with spinach and cabbage, lettuce had a relatively large specific surface area, resulting in a higher photosynthetic rate, and thus the highest *δ*^13^C value. Although cabbage had a relatively small specific surface area due to its wrapping structure, it had a longer growth period than lettuce and spinach, resulting in its *δ*^13^C value (−29.6 ± 0.4‰) between lettuce and spinach. There were significant differences in δ^15^N values among the three vegetables (*p* < 0.05), and the order was lettuce > spinach > cabbage. Lettuce had the highest *δ*^15^N value (13.0 ± 0.1‰), although spinach and lettuce were applied with the same base fertilizer only once, indicating that the nitrogen use efficiency of the three vegetables differed [29]. Cabbage had the lowest *δ*^15^N value (4.6 ± 0.3‰), possibly due to the application of a chemical compound fertilizer with a low *δ*^15^N value (−1.2‰) during its growing period [7,10,18]. There were significant differences in *δ*^2^H values among the three vegetables (*p* < 0.05). Cabbage had the highest *δ*^2^H value (−72.2 ± 0.3‰), followed by lettuce (−86.0 ± 0.1‰), and spinach had the lowest (−113.4 ± 0.2‰). This might be because ^1^H_2_O more easily diffused from leaves to the atmosphere during transpiration, thus resulting in ^2^H enrichment in leaves [7]. Cabbage had a longer growth period and longer transpiration time compared to lettuce and spinach, while lettuce had a larger specific surface area and thus a higher transpiration per unit time. The *δ*^18^O and *δ*^2^H values of atmospheric precipitation were positively correlated (*δ*^2^H = 8*δ*^18^O + 10), but the relative mass difference between ^18^O and ^16^O was much smaller than between ^2^H and ^1^H, and therefore the fractionation of the oxygen isotope was less than that of the hydrogen isotope [7,11]. Moreover, oxygen, as a component of CO_2_, was also affected by plant photosynthesis [7]. Under the influence of multiple factors, cabbage had the highest *δ*^18^O value, which was significantly different from spinach and lettuce. These results illustrated that different types of vegetables exhibited different stable isotope ratios due to their physiological and biochemical differences even under the same climatic and environmental conditions, even with the application of the same or similar fertilizers.

The number of applications of thiamethoxam and fenvalerate had different effects on the change in *δ*^13^C, *δ*^15^N, *δ*^2^H, and *δ*^18^O values in spinach, cabbage, and lettuce (Table 2). The *δ*^13^C values of spinach were significantly different between thiamethoxam applied twice and the control group (*p* < 0.05), and spinach with the first application of thiamethoxam had the highest *δ*^13^C value (−28.5‰), which could be due to the relatively high *δ*^13^C value (−18.0‰) of thiamethoxam, and because thiamethoxam had a good internal absorption conductivity and was rapidly transmitted to all parts of the plant [19]. The *δ*^13^C value (−29.5‰) of the second application was more negative, which might be attributed to the higher concentration of the second application that blocked the leaf stomatas [30]. The *δ*^15^N values of spinach after two thiamethoxam applications were higher than those of the control group and significantly different from the control group (*p* < 0.05), although the δ^15^N value of thiamethoxam was −0.60‰. This could be attributed to the enrichment of ^15^N in the process of thiamethoxam metabolism to produce clothianidin, nitroso compounds, guanidine, urea, and other compounds under the action of enzymes [20,25,31]. The *δ*^2^H value of spinach with the first application of thiamethoxam was slightly lower than that of the control group, but the difference was not significant (*p* > 0.05), and the *δ*^2^H value after the second application was significantly higher than that of the first and the control group. This might be attributed to the higher *δ*^2^H value of thiamethoxam (−8.4‰); however, the *δ*^2^H value of the plant was mainly affected by the water source and transpiration, and the high concentration of pesticide droplets might block stomata on the leaf surface and thus affected water transpiration [7,30]. The *δ*^18^O value of spinach with the first application of thiamethoxam was the highest, followed by that of the second application, but only the δ^18^O value of the first application was significantly different from the control group (*p* < 0.05), which might be attributed to the high *δ*^18^O value of thiamethoxam (31.4‰), and the fact that oxygen is a component of water and carbon dioxide [7,11], and changes in photosynthetic efficiency and water use efficiency caused by blocking stomata with pesticide droplets could also affect the oxygen isotope ratio of spinach. After the two applications of fenvalerate, the control group had the lowest *δ*^13^C value and the difference was significant compared with the first and the second application of fenvalerate, which may be attributed to the relatively high *δ*^13^C value of fenvalerate (−27.1‰); however, fenvalerate is a lipid-soluble pesticide and more easily attache to the leaf surface [21,22], and it could affect the *δ*^13^C value of spinach by influencing the photosynthesis of leaves. The *δ*^15^N, *δ*^2^H, and *δ*^18^O values in spinach showed no significant difference among the first and second application of fenvalerate and the control group, which might be mainly attributed to the fact that fenvalerate was a lipid-soluble pesticide and its permeability in spinach leaves was relatively low [21,32,33].

After the two applications of thiamethoxam in cabbage, only the *δ*^2^H values of thiamethoxam were significantly different between the control group and the two applications (*p* < 0.05), and the control group had the highest δ^2^H values. This might be attributed to the fact that thiamethoxam was basically sprayed in the outer layer of cabbage (Figure 2), which might affect water use efficiency by blocking the leaf stomata of cabbage, and thus affected *δ*^2^H values. After the two applications of fenvalerate, the *δ*^13^C, *δ*^2^H, and *δ*^18^O values of cabbage were relatively low, and all of them showed significant differences from the control group (*p* < 0.05). However, the stable isotope ratio differences between the two applications were not significant, which could be related to the lipid-soluble and the lower *δ*^2^H (−85.8‰) and *δ*^18^O (14.8‰) of fenvalerate, and the wrapping structure and the waxy cuticle of cabbage. Although fenvalerate was mostly attached to the outer surface of cabbage, the permeability of cabbage leaves could be relatively higher than that of spinach leaves due to the thicker waxy cuticle of cabbage leaves [34]. This also explained why there were significant differences in the stable isotope ratios of cabbage between applications of fenvalerate and the control group, while there was no significant difference in *δ*^15^N, *δ*^2^H, or *δ*^18^O values between applications of fenvalerate and the control group. The *δ*^15^N values of cabbage with fenvalerate (*δ*^15^N = −1.4‰) applied were higher than that of the control group (*p* < 0.05), which might be mainly attributed to the interaction of fenvalerate with enzymes, organic acids, and polyphenols in cabbage [21,35], which enriched ^15^N.

The change in the *δ*^13^C value in lettuce with thiamethoxam applied was the same as that of spinach, in which the value first increased and then decreased. There was a significant difference between the two applications, but no significant difference compared with the control. This might be mainly due to the good internal absorption of thiamethoxam and the high concentration of thiamethoxam that blocked the stomata of lettuce leaves [19,36]. The *δ*^15^N values of lettuce with thiamethoxam residue were significantly lower than those of the control group (*p* < 0.05). This was inconsistent with the change in spinach and cabbage, and it was ascribed to the high residue concentrations of thiamethoxam in lettuce (Figure 2) and the low *δ*^15^N value (−0.6‰), although ^15^N could be enriched in the metabolic process of thiamethoxam in lettuce. The *δ*^2^H value of lettuce increased gradually from the control group to the two applications of thiamethoxam, and the *δ*^2^H value of lettuce after the second application was significantly different from that of the control group (*p* < 0.05), which was mainly attributed to the high thiamethoxam residue levels and thiamethoxam having a high *δ*^2^H value (−8.4‰). There were no significant differences in the *δ*^18^O values of lettuce among the different applications of thiamethoxam due to the combined effects of the *δ*^18^O values of thiamethoxam, leaf photosynthetic efficiency and water use efficiency [8,11]. The trend in *δ*^13^C values in lettuce after applying fenvalerate was the same as that after applying thiamethoxam, and the *δ*^13^C value of lettuce after the first application of fenvalerate was the highest and significantly different than that of the second application. This could be ascribed to the fact that the waxy cuticle on the surface of the lettuce leaves could increase the permeability of fenvalerate and the fenvalerate inside of the lettuce could reduce carbohydrate metabolism [35], while the high concentration of fenvalerate could also block the stomata of lettuce leaves and affect photosynthetic efficiency [34], leading to the *δ*^13^C value increasing first and then decreasing. The effect of fenvalerate on the *δ*^15^N values of lettuce was also the same as that of thiamethoxam. The *δ*^15^N values of lettuce with fenvalerate applied were significantly lower than the control group, which could also be attributed to the high fenvalerate residue in lettuce, the waxy cuticle on the lettuce surface increasing the permeability of fenvalerate and fenvalerate having a low *δ*^15^N value (−1.4‰) [34]. There was no significant difference in the *δ*^2^H values of lettuce with different application (one or two) of fenvalerate, although the *δ*^2^H values increased slightly after two applications compared with the control group, which was consistent with fenvalerate applied to spinach, and might be attributed to fenvalerate attaching to the surface of lettuce, blocking the leaf stomata, and affecting the water use efficiency of lettuce [21,22,34]. Lettuce with fenvalerate applications had higher *δ*^18^O values, and the *δ*^18^O value of lettuce after the first fenvalerate application was the highest, which was significantly different from the control group. This might also be due to fenvalerate affecting the photosynthetic efficiency and water use efficiency of lettuce leaves, which thus enriched ^18^O, although fenvalerate could penetrate into lettuce through the waxy cuticle and had a relatively low *δ*^18^O value (14.8‰). These results indicated that the structure and residue of thiamethoxam and fenvalerate, the leaf structure of the vegetables, and the metabolic processes of the pesticides jointly influenced the stable isotope distribution in vegetables. Therefore, it is of great significance to further analyze the changes in stable isotopes during pesticide metabolism and the relationship between pesticide residue levels and stable isotope ratios for identifying the cultivation pattern of vegetables through stable isotope techniques.

### 3.3. Relationship between Stable Isotopes and Pesticide Residues

The degradation processes of thiamethoxam and fenvalerate in spinach, cabbage, and lettuce were different over time, and the changes in stable isotope ratios were also different, partly due to the differences in pesticide residues. Pearson correlation coefficient method was used to analyze the relationship between the *δ*^13^C, *δ*^15^N, *δ*^2^H, and *δ*^18^O values of the three vegetables and the residue levels of thiamethoxam and fenvalerate after the pesticides were applied twice (Table 3). The *δ*^13^C and *δ*^15^N values of spinach after the first application of thiamethoxam were significantly positively correlated with the residual amount (*p* < 0.05), and the correlation coefficients were 0.9046 and 0.8591, respectively, indicating that the changes in the *δ*^13^C and *δ*^15^N values were consistent with the degradation metabolism of thiamethoxam residue in spinach (Figure 2a). This might be mainly attributed to the good internal absorption conductivity of thiamethoxam [19], with thiamethoxam having a high *δ*^13^C value (−18.02‰) and low *δ*^15^N value (−0.60‰). Thiamethoxam entering spinach could be gradually metabolized under the action of enzymes over time to produce thiamethoxam and other products [19,20] and the *δ*^13^C value could show a decreasing trend during this process, while the δ^15^N value might show an increasing trend due to the enrichment of ^15^N. The *δ*^18^O values of spinach showed a strong positive correlation with pesticide residue levels (r = 0.6507), and the *δ*^2^H values showed a weak negative correlation with residue levels (r = −0.3059), showing that the trend of the *δ*^18^O values was consistent with that of thiamethoxam residue levels, while the trend of the *δ*^2^H values was opposite to that of thiamethoxam residue levels, although the correlations were not statistically significant (*p* > 0.05). This might be attributed to the influence of thiamethoxam on the surface stomata of the spinach leaf and thus on its water transpiration and photosynthesis [22,34]. After the second application of thiamethoxam in spinach, the change in the *δ*^13^C value was not significantly positively correlated with the thiamethoxam residue level, and the changes in *δ*^15^N, *δ*^2^H, and *δ*^18^O values were not significantly negatively correlated with the thiamethoxam residue level. This might be mainly due to the dual effects of high concentrations of thiamethoxam. Thiamethoxam blocked the surface stomata on spinach leaves, resulting in changes in photosynthetic efficiency and water use efficiency [36]; it also affected the metabolism of thiamethoxam in vivo (Figure 2b and Table 1) and the enrichment of ^15^N. After the first application of fenvalerate, the *δ*^13^C, *δ*^15^N, *δ*^2^H, and *δ*^18^O values of spinach were strongly positively correlated with fenvalerate residue level, but only the correlation of the *δ*^13^C value (r = 0.9514) was statistically significant (*p* < 0.05), which might be mainly attributed to the fact that fenvalerate was a lipid-soluble pesticide that mainly adhered to the leaf surface, and its metabolism could be affected by the external light intensity, temperature, and humidity, which was consistent with the trend of the stable isotope ratios caused by photosynthesis, transpiration, and respiration [21,34]. After the second application of fenvalerate in spinach, the *δ*^13^C, *δ*^15^N, and *δ*^2^H values were not significantly correlated with fenvalerate residue levels, while the *δ*^18^O value was not significantly positively correlated with fenvalerate residue levels. This could be due to some of the fenvalerate permeating into spinach through the waxy cuticle where it influenced carbohydrate metabolism, while most of it adhered to the surface of spinach leaves where it affected photosynthetic efficiency and water use efficiency [34,37].

The *δ*^13^C, *δ*^15^N, *δ*^2^H, and *δ*^18^O values of the two thiamethoxam applications in cabbage showed weak or moderate correlations with the thiamethoxam residue levels, but the correlations were statistically insignificant (*p* > 0.05), which might be mainly related to the wrapping structure of cabbage with thiamethoxam mainly sprayed on the outer leaves of the cabbage. After the first application of fenvalerate, the *δ*^13^C (r = −0.7600) and *δ*^2^H values (r = −0.8458) showed significantly strong negative correlations with the fenvalerate residue levels of cabbage, showing that the trend of *δ*^13^C and *δ*^2^H values was opposite to that of fenvalerate residue levels, which might be due to fenvalerate entering the interior of the cabbage through its waxy cuticle affecting carbohydrate metabolism, and fenvalerate on the surface of the cabbage leaves affecting water use efficiency [8,37]. The *δ*^15^N value (r = 0.8659) showed a significantly strong positive correlation with the pesticide residue levels, indicating that the trend of the *δ*^15^N value was consistent with that of fenvalerate residue levels, which might be attributed to the decrease in the interaction between fenvalerate and chemical components (enzymes, organic acids, and polyphenols) in cabbage with the degradation of fenvalerate. The *δ*^18^O value was not significantly negatively correlated with the pesticide residue level, due to the comprehensive effect of fenvalerate on photosynthetic efficiency and water use efficiency of cabbage [8,34]. After the second application of fenvalerate, only the *δ*^13^C value (r = 0.9471) showed a strong and significant positive correlation with fenvalerate residue, while the other stable isotope rations were weakly negatively correlated with pesticide residue levels, which might be mainly attributed to the effect on photosynthetic efficiency of high concentrations of fenvalerate on the outer leaf surface of cabbage, and the effect of fenvalerate penetrating into cabbage on carbohydrate metabolism.

After the first application of thiamethoxam in lettuce, the *δ*^13^C value showed a strong positive correlation with the thiamethoxam residue level, although the relationship was not statistically significant, which indicated that the trend of the *δ*^13^C value was consistent with the thiamethoxam residue levels. This might be attributed to thiamethoxam having a high *δ*^13^C value and being metabolized in lettuce over time. There was a significantly strong negative correlation between the *δ*^15^N value and thiamethoxam residue level (r = −0.94259), which was different from the first application of thiamethoxam in spinach, and might be attributed to the enrichment rate of ^15^N in thiamethoxam metabolism exceeding the influence of the *δ*^15^N value (−0.60‰) of thiamethoxam itself (Figure 2a). The *δ*^2^H (r = 0.8777) and *δ*^18^O values (r = 0.9011) were significantly positively correlated with thiamethoxam residue levels, although thiamethoxam had the higher *δ*^2^H (−8.4‰) and *δ*^18^O values (31.4‰), which might be attributed to the fact that the metabolism of thiamethoxam required the participation of water [8,11]. The *δ*^2^H (−28.53‰) and *δ*^18^O (−5.10‰) values of irrigation water were relatively low, resulting in a trend of *δ*^2^H and *δ*^18^O values consistent with that of the thiamethoxam residue levels. Only the *δ*^15^N value showed a significant negative correlation with the thiamethoxam residue level (r = −0.9612) of lettuce after the second application of thiamethoxam, while the *δ*^13^C, *δ*^2^H, and *δ*^18^O values showed no significant correlation with the pesticide residue level, which might be mainly attributed to the high concentration of thiamethoxam attached to the leaf surface affecting photosynthesis, respiration and water use efficiency [22,26,34]. The *δ*^13^C (r = 0.8313) and *δ*^18^O (r = 0.9593) values of lettuce after the first application of fenvalerate showed a significant and strong positive correlation with fenvalerate residue levels, while the *δ*^15^N and *δ*^2^H values were not significantly correlated with fenvalerate residue levels. This might be mainly attributed to the fact that most of the fenvalerate was attached to the surface of the lettuce leaves, and the metabolism of fenvalerate under the action of external factors was consistent with the trend of stable isotope ratios caused by photosynthesis, transpiration, and respiration. Furthermore, some of the fenvalerate entered the interior of lettuce through the waxy cuticle and affected the metabolism of carbohydrates [35]. After the second application of fenvalerate, the *δ*^13^C, *δ*^15^N, *δ*^2^H, and *δ*^18^O values of lettuce showed a negative correlation with the thiamethoxam residue levels, which were not statistically significant and might be attributed to a high concentration of fenvalerate blocking the stomata of lettuce leaves, affecting the photosynthetic efficiency and water use efficiency of lettuce, slowing its metabolism inside and outside of the lettuce (Figure 2b). These results indicated that the change in the stable isotope ratios was affected by pesticide residues in vegetables, and the relationship between stable isotope ratios and pesticide residue levels was mainly determined by the metabolic processes of pesticides, the photosynthetic efficiency, and water use efficiency of the three vegetables, which were influenced by pesticide structure, leaf surface characteristics (specific surface area, stomata, and waxy cuticle), and the growth stage of the three vegetables.

### 3.4. Identification of Vegetables with Pesticide Residues

The stable isotope ratios of the three vegetables were influenced by different pesticides and the times they were applied. Spinach, cabbage, and lettuce with different application (one or two) of thiamethoxam and fenvalerate were identified using *δ*^13^C, *δ*^15^N, *δ*^2^H, and *δ*^18^O values and the PLS-DA method [38]. The three vegetables were grouped separately on the first two principal component score plots of PLS-DA, for which the identification accuracies were all 100% (Figure 4 and Table 4). This might be mainly attributed to the different physiological and biochemical metabolism of the three vegetables, although the cultivation conditions were similar. The control groups of the three vegetables (Figure 4, in the elliptic circle) were also significantly distinguished from the vegetables containing pesticide residues, which might be mainly attributed to the effect of different concentrations of thiamethoxam and fenvalerate on the *δ*^13^C, *δ*^15^N, *δ*^2^H, and *δ*^18^O values in spinach, cabbage and lettuce. The variable importance of projection [38] (VIP, indicating the isotope that had the most influence on geographic discrimination of the three vegetables, see Figure 4b) was *δ*^2^H > *δ*^13^C > 1 > *δ*^15^N > *δ*^18^O, which showed that the *δ*^2^H and *δ*^13^C values, mainly influenced by transpiration, pesticide residues, plant respiration and photosynthesis, respectively, were the most important factors used to distinguish spinach, cabbage, and lettuce from the same growing region.

The accuracies of the PLS-DA model to identify the control group, the first and second application of thiamethoxam on spinach (spinach-thiamethoxam-first and spinach-thiamethoxam-second for short, respectively, and the same for cabbage and lettuce) were 100%, and the identification accuracies of the first and second application of fenvalerate samples (spinach-fenvalerate-first and spinach-fenvalerate-second for short, respectively, and the same for cabbage and lettuce) were 75% (Table 4), with one spinach-fenvalerate-first sample being misclassified as a spinach-thiamethoxam-first sample, and one spinach-fenvalerate-second sample being misclassified as a spinach-fenvalerate-first sample. The results indicated whether or not the application of pesticide and the type of pesticide affected the stable isotope ratios of spinach, which then affected the discrimination accuracy of the PLS-DA model. The order of VIP was *δ*^18^O > *δ*^15^N > 1 > *δ*^13^C > *δ*^2^H, suggesting that the *δ*^18^O value affected by the water use efficiency and photosynthetic efficiency, and the *δ*^15^N value possibly affected by thiamethoxam metabolism in vegetables were the two most important variables influencing the identification of spinach with different concentrations of pesticides.

The identification accuracies of the cabbage control group, cabbage-thiamethoxam-second, and cabbage-fenvalerate-first were 100%, and that of cabbage- fenvalerate-second was 75%, while the cabbage-thiamethoxam-first was 50%, which might be related to the wrapping structure and waxy cuticle of cabbage and the properties of thiamethoxam and fenvalerate [19,21,34]. After the first application, the wrapping structure of cabbage retained thiamethoxam and fenvalerate mostly on the outer leaf, and the good internal absorption conductivity of thiamethoxam enabled it to quickly enter the inside of the outer leaf for metabolism, while only some of fenvalerate could enter the inside of the outer leaf of cabbage through the waxy cuticle, and the distribution of the pesticide in cabbage was in an unbalanced state. The metabolism of thiamethoxam inside of cabbage mainly affected the *δ*^15^N value [31], while thiamethoxam and fenvalerate attached to the leaf surface mainly affected the *δ*^13^C, *δ*^2^H, and *δ*^18^O values by influencing the photosynthetic efficiency and water use efficiency of cabbage [8,11,30], resulting in one cabbage-thiamethoxam-first sample being wrongly identified as cabbage-thiamethoxam-second and one as spinach-fenvalerate-first. The pesticide concentration was doubled at the second application, and the pesticide residues showed an accumulation phenomenon, especially for fenvalerate, and more pesticides might have been attached to the outer layer for the cabbage. After distribution and metabolism of the first application, the stable isotope ratios at the early stage of the second application were potentially similar to those at the late stage of the first application, leading to the spinach-fenvalerate-second sample being incorrectly classified as spinach-fenvalerate-first. According to the order of the VIP, namely *δ*^15^N > *δ*^18^O > 1 > *δ*^13^C > *δ*^2^H, the *δ*^15^N and *δ*^18^O values were the two most important variables affecting the identification accuracy of cabbage, and might be due to the properties of thiamethoxam and fenvalerate, and the wrapping structure and waxy cuticle of cabbage [19,21,30,34].

The identification accuracies of the PLS-DA model for the control group of lettuce and lettuce-thiamethoxam-first were 100%, and the accuracies of lettuce-thiamethoxam-second, lettuce-fenvalerate-first, and lettuce-fenvalerate-second were 75%, 25% and 50%, respectively. This might be mainly attributed to the effect of different amounts of pesticide residues (Figure 2) and different metabolic processes of thiamethoxam and fenvalerate on stable isotopes of lettuce. There was one lettuce-thiamethoxam-second sample misclassified as lettuce-fenvalerate-second, three lettuce-fenvalerate-first samples misclassified as lettuce-thiamethoxam-first, and two lettuce-fenvalerate-second samples misclassified as lettuce-thiamethoxam-second. This could be due to thiamethoxam and fenvalerate entering the lettuce interior affecting nitrogen and carbon stable isotopes [31,35], while pesticides attached to the leaf surface affected photosynthetic efficiency and water use efficiency of lettuce, and thus affected carbon, hydrogen, and oxygen stable isotopes [8,11,30]. The comprehensive effect of these factors made the distribution of stable isotope ratios of thiamethoxam and fenvalerate similar among the different numbers of applications in lettuce. According to the order of VIP (*δ*^13^C > *δ*^18^O > 1 > *δ*^15^N > *δ*^2^H), the most important variables were *δ*^13^C and *δ*^18^O values, which were also consistent with the change in stable isotopes caused by the metabolism of pesticides and photosynthesis, respiration, and transpiration of lettuce.

According to these results, the control groups of the three vegetables were completely identified, and the total accuracy of spinach was 88.89%, cabbage was 83.33%, and lettuce was 66.67%, which were a result of the effects of the stable isotopes of the pesticides themselves, the structure of the three vegetables and the physiological and biochemical metabolism of vegetables affected by thiamethoxam and fenvalerate. Although the identification accuracies among different types of pesticides and different times of application in the three vegetables still need to be further improved, especially for lettuce, these results confirm the effects of pesticide residues on stable isotopes and indicate that the identification of different pesticide residues can be performed based on stable isotope technology.

## 4. Conclusions

The effects of thiamethoxam and fenvalerate residues on light-stable isotope ratios (*δ*^13^C, *δ*^15^N, *δ*^2^H, and *δ*^18^O) in spinach, cabbage, and lettuce were investigated. The residue levels and degradation processes of thiamethoxam and fenvalerate after two applications on the three vegetables differed. The *δ*^13^C, *δ*^15^N, *δ*^2^H, and *δ*^18^O values of the three vegetables in the control group were significantly different. There were significant differences in the stable isotope ratios among different applications (one or two) of thiamethoxam in the three vegetables except for the *δ*^13^C, *δ*^15^N, and *δ*^18^O values of cabbage and the *δ*^18^O values of lettuce. After fenvalerate application, the *δ*^15^N, *δ*^2^H, and *δ*^18^O values of spinach and the *δ*^2^H value of lettuce were not significantly different based on number of applications, while there were significant differences in other stable isotope ratios after different numbers of applications on the three vegetables. The residue levels of thiamethoxam and fenvalerate not only affected the light-stable isotopes distribution in the three vegetables, but they also had a certain correlation with the change in the stable isotope ratios. There was a 54.17% stable isotope ratio that significantly correlated with pesticide residue levels. Based on the difference in stable isotope distributions of vegetables after thiamethoxam and fenvalerate application, the total identification accuracy of the PLS-DA model for spinach was 88.89%, cabbage was 83.33%, and lettuce was 66.67%. The important variables affecting the identification of spinach and cabbage were *δ*^15^N and *δ*^18^O values, and the variables affecting the identification of lettuce were *δ*^13^C and *δ*^18^O values. These results suggest that the stable isotope technique combined with multivariate statistics and chemometrics methods can identify the thiamethoxam and fenvalerate residues of spinach, cabbage, and lettuce, and can provide reference methods for the identification of vegetables with pesticide residue. However, it is necessary to further increase the research on the changes in stable isotopes by applying pesticides at the different growth stages of vegetables, and increase the types and number of vegetables to improve the universality and robustness of the discriminant model of vegetables containing different pesticide residues.

## Figures and Tables

**Figure 1 foods-12-02655-f001:**
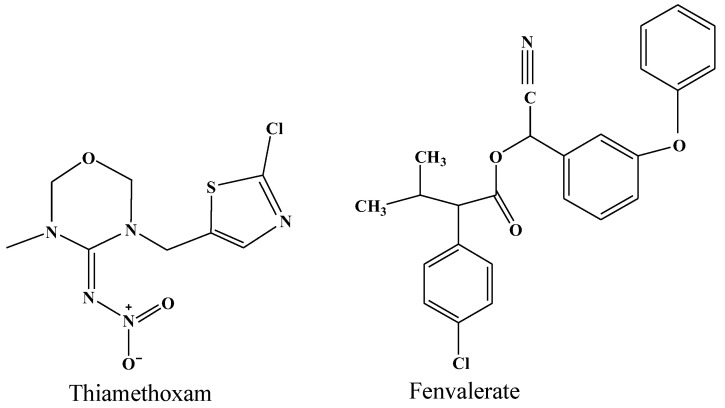
Structural formulas of thiamethoxam and fenvalerate.

**Figure 2 foods-12-02655-f002:**
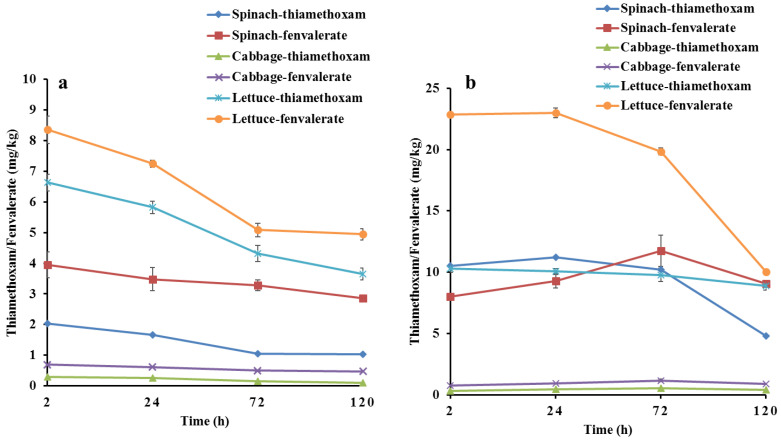
Degradation curve of thiamethoxam and fenvalerate for the first (**a**) and the second (**b**) applications in spinach, cabbage, and lettuce.

**Figure 3 foods-12-02655-f003:**
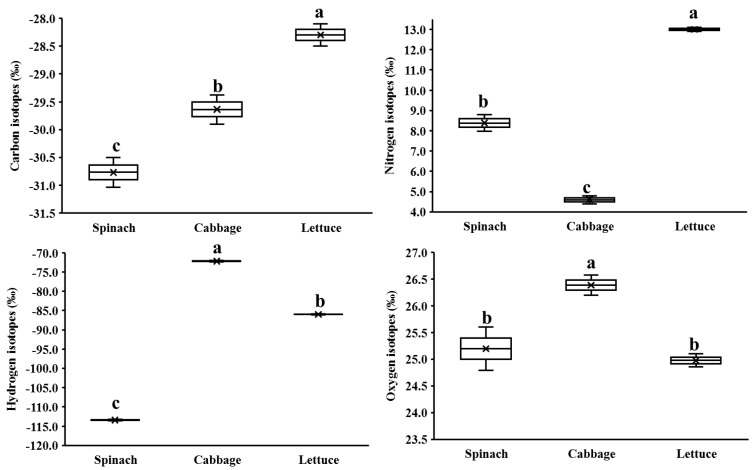
Distribution of stable isotopes in the three vegetable control groups. “×” represents mean values; “-” represents median values; different letters within a row indicate a significant difference for each region (*p* < 0.05).

**Figure 4 foods-12-02655-f004:**
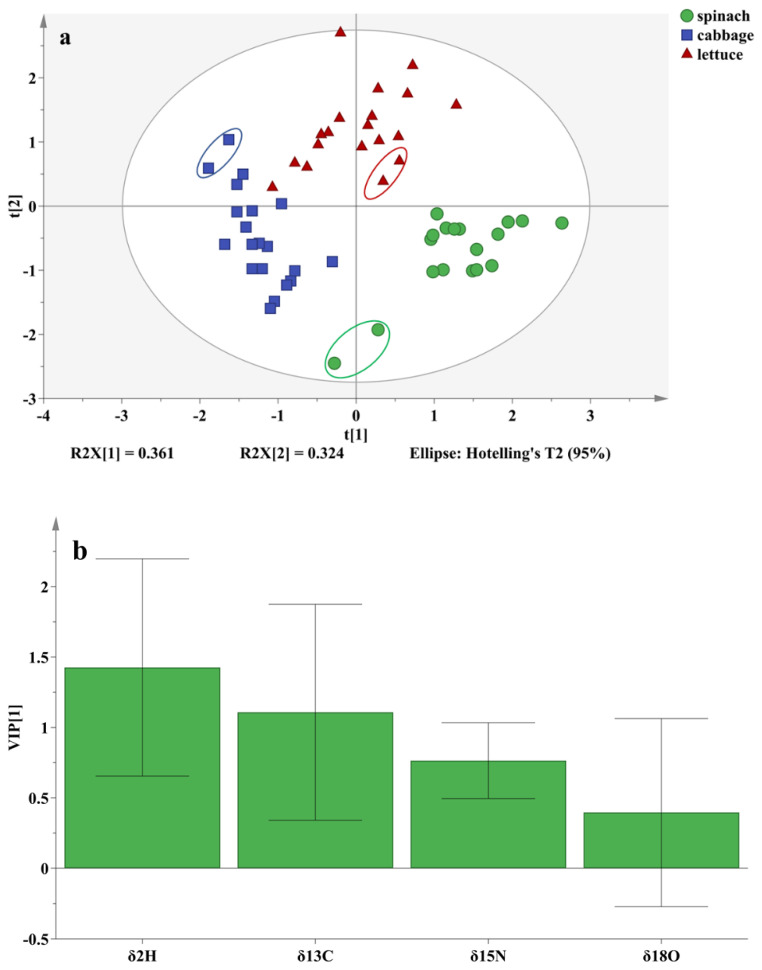
PLS-DA score plots (**a**) and VIPs (**b**) for spinach, cabbage, and lettuce with different pesticide residue levels.

**Table 1 foods-12-02655-t001:** Degradation equations of thiamethoxam and fenvalerate.

Vegetables	Pesticides	Degradation Equation	Correlation Coefficient	Residual Standard Deviation	Degradation Half-Life/d
spinach	thiamethoxam	$y=1.9241e^{-0.0060x}$	0.9520 *	0.1837	115.76
	fenvalerate	$y=3.8594e^{-0.0025x}$	0.9678 *	0.1396	279.64
cabbage	thiamethoxam	$y=0.3070e^{-0.0094x}$	0.9873 *	0.0178	73.60
	fenvalerate	$y=0.6682e^{-0.0032x}$	0.9661 *	0.0317	216.09
lettuce	thiamethoxam	$y=6.5756e^{-0.0051x}$	0.9933 **	0.1921	134.90
	fenvalerate	$y=8.0589e^{-0.0047x}$	0.9568 *	0.5959	148.24

Note: “*” means significant; “**” means extremely significant.

**Table 2 foods-12-02655-t002:** Differences in the stable isotope ratios with one or two applications of pesticides on leafy vegetables.

Vegetables	Pesticides	Application of Pesticides	*δ^1^*^3^C/‰	*δ*^15^N/‰	*δ*^2^H/‰	*δ*^18^O/‰
Spinach	Thiamethoxam	Control	−30.8 ± 0.38 c	8.4 ± 0.59 b	−113.4 ± 0.24 b	25.2 ± 0.57 b
		The first	−28.5 ± 0.29 a	12.0 ± 0.99 a	−114.6 ± 2.58 b	27.3 ± 0.56 a
		The second	−29.5 ± 0.25 b	13.3 ± 1.65 a	−111.2 ± 0.76 a	26.5 ± 0.56 ab
	Fenvalerate	Control	−30.8 ± 0.38 b	8.4 ± 0.59 a	−113.4 ± 0.24 a	25.2 ± 0.57 a
		The first	−28.6 ± 0.12 a	10.0 ± 1.00 a	−108.0 ± 4.58 a	25.9 ± 0.30 a
		The second	−28.8 ± 0.22 a	9.2 ± 0.25 a	−110.1 ± 1.91 a	26.2 ± 0.83 a
Cabbage	Thiamethoxam	Control	−29.6 ± 0.37 a	4.6 ± 0.29 a	−72.2 ± 0.33 a	26.4 ± 0.27 a
		The first	−30.2 ± 0.46 a	6.1 ± 2.07 a	−83.7 ± 3.87 b	26.4 ± 0.83 a
		The second	−30.5 ± 0.44 a	6.0 ± 1.82 a	−85.7 ± 6.29 b	26.7 ± 0.25 a
	Fenvalerate	Control	−29.6 ± 0.37 a	4.6 ± 0.29 b	−72.2 ± 0.33 a	26.4 ± 0.27 a
		The first	−30.6 ± 0.27 b	10.4 ± 0.88 a	−86.2 ± 7.49 b	25.1 ± 0.66 b
		The second	−31.1 ± 0.28 b	10.5 ± 0.13 a	−86.6 ± 6.70 b	25.3 ± 0.28 b
Lettuce	Thiamethoxam	Control	−28.3 ± 0.28 ab	13.0 ± 0.14 a	−86.0 ± 0.14 b	25.0 ± 0.17 a
		The first	−27.8 ± 0.36 a	8.1 ± 0.41 b	−83.3 ± 2.14 ab	27.0 ± 1.2 a
		The second	−28.9 ± 0.25 b	8.7 ± 2.10 b	−81.0 ± 1.04 a	25.9 ± 1.1 a
	Fenvalerate	Control	−28.3 ± 0.28 ab	13.0 ± 0.1 a	−86.0 ± 0.14 a	25.0 ± 0.2 b
		The first	−27.9 ± 0.11 a	8.1 ± 1.9 b	−78.8 ± 8.07 a	27.5 ± 1.0 a
		The second	−28.5 ± 0.26 b	8.6 ± 1.30 b	−80.2 ± 4.54 a	26.6 ± 0.7 ab

Note: Different lowercase letters within a column indicate significant difference among one or two applications of pesticides at the *p* < 0.05 level.

**Table 3 foods-12-02655-t003:** Relationship between stable isotope ratios and thiamethoxam and fenvalerate residues in leafy vegetables.

Vegetable Varieties	Pesticides	Stable Isotope Ratios/‰	Application Times of Pesticides	r
Spinach	Thiamethoxam	*δ*^13^C	The first	**0.9046 ***
			The second	0.2680
		*δ*^15^N	The first	**0.8591 ***
			The second	−0.6405
		*δ*^2^H	The first	−0.3059
			The second	−0.3399
		*δ*^18^O	The first	0.6507
			The second	−0.4987
	Fenvalerate	*δ*^13^C	The first	**0.9514 ***
			The second	−0.2860
		*δ*^15^N	The first	0.7707
			The second	−0.6623
		*δ*^2^H	The first	0.7310
			The second	−0.3613
		*δ*^18^O	The first	0.7140
			The second	0.7774
Cabbage	Thiamethoxam	*δ*^13^C	The first	−0.2364
			The second	0.1117
		*δ*^15^N	The first	0.1659
			The second	−0.1603
		*δ*^2^H	The first	−0.5354
			The second	0.1499
		*δ*^18^O	The first	0.4860
			The second	0.5646
	Fenvalerate	*δ*^13^C	The first	**−0.7600 ***
			The second	**0.9471 ***
		*δ*^15^N	The first	**0.8659 ***
			The second	−0.3230
		*δ*^2^H	The first	**−0.8458 ***
			The second	−0.2824
		δ^18^O	The first	−0.5580
			The second	−0.0143
Lettuce	Thiamethoxam	*δ*^13^C	The first	0.7535
			The second	0.2848
		*δ*^15^N	The first	**−0.9426 ***
			The second	**−0.9612 ***
		*δ*^2^H	The first	**0.8777 ***
			The second	0.3346
		*δ*^18^O	The first	**0.9011 ***
			The second	−0.6327
	Fenvalerate	*δ*^13^C	The first	**0.8314 ***
			The second	−0.3535
		*δ*^15^N	The first	−0.7384
			The second	0.2045
		*δ*^2^H	The first	0.5500
			The second	−0.3621
		*δ*^18^O	The first	**0.9594 ***
			The second	−0.5497

Note: The asterisk (*) indicates a significant correlation between a stable isotope ratio and pesticide residue at the *p* < 0.05 level.

**Table 4 foods-12-02655-t004:** PLS-DA model performances of vegetables with different concentrations of pesticides.

Vegetable Varieties	Predictive Accuracy (%)	Total Accuracy (%)	The Order of VIP
Spinach-control	100	88.89	*δ*^18^O > *δ*^15^N > 1 > *δ*^13^C > *δ*^2^H
Spinach-thiamethoxam-first	100		
Spinach-thiamethoxam-second	100		
Spinach-fenvalerate-first	75		
Spinach-fenvalerate-second	75		
Cabbage-control	100	83.33	*δ*^15^N > *δ*^18^O > 1 > *δ*^13^C > *δ*^2^H
Cabbage-thiamethoxam-first	50		
Cabbage-thiamethoxam-second	100		
Cabbage-fenvalerate-first	100		
Cabbage-fenvalerate-second	75		
Lettuce-control	100	66.67	*δ*^13^C > *δ*^18^O > 1 > *δ*^15^N > *δ*^2^H
Lettuce-thiamethoxam-first	100		
Lettuce-thiamethoxam-second	75		
Lettuce-fenvalerate-first	25		
Lettuce-fenvalerate-second	50		

## Data Availability

Data is contained within the article.

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
