# Peer review of "Effects of Thiamethoxam and Fenvalerate Residue Levels on Light-Stable Isotopes of Leafy Vegetables"

_foods, 2023, doi:10.3390/foods12142655_

Round 1

Reviewer 1 Report

The manuscript titled "Effects of Thiamethoxam and Fenvalerate Residue Levels on Light Stable Isotopes of Leafy Vegetable" by  Qi et al. aims to investigate the relationship between different insecticide residues and light stable isotope ratios by continuous application of thiamethoxam and fenvalerate on the common leaf vegetables cabbage, lettuce, and spinach.

The following comments need to be addressed in the revised version:

- Please revise the first sentence of introduction for some mistakes.

- Please put the explanation in lines 53-54 of Green in an independent sentence.

- Please replace "Green and organic agricultural" in lines 58-59 by "They". 

- please transfer the structural formula of both insecticides to materials and methods under chemicals and reagents subtitle. Also delete what mentioned in the last sentence of introduction.

- Under 2.3. sample preparation: Please provide the method of extraction and clean up used for preparing the samples for residue analysis.

- Please mention the name of analytical device and its specifications under point 2.5. Thiamethoxam and Fenvalerate Residue Determination.

- Conclusion is very long. Please make it short and concise to give a very specific information about major findings and their effects.

- Also please revise the whole manuscript for some grammatical and language errors.

Moderate editing of English language required.

Author Response

Reviewer 1:

Comments and Suggestions for Authors

The manuscript titled “Effects of Thiamethoxam and Fenvalerate Residue Levels on Light Stable Isotopes of Leafy Vegetable” by Qi et al. aims to investigate the relationship between different insecticide residues and light stable isotope ratios by continuous application of thiamethoxam and fenvalerate on the common leaf vegetables cabbage, lettuce, and spinach.

The following comments need to be addressed in the revised version:

- Please revise the first sentence of introduction for some mistakes.

Response: Thank you for the comments. The first sentence of introduction has been revised as follows “The rational use of chemical pesticides can ensure agricultural products output and improve agricultural economic benefits, but frequent and improper use not only leads to crops and pests developing resistance, but also it also leads to excessive residues in crops and environmental pollution.”, which was showed in lines 42-45 of the revised manuscript.

- Please put the explanation in lines 53-54 of Green in an independent sentence.

Response: We have explained the green product in an independent sentence, showed in lines 56-59 of the revised manuscript.

- Please replace "Green and organic agricultural" in lines 58-59 by "They".

Response: We have replaced the “Green and organic agricultural” as “They”.

- please transfer the structural formula of both insecticides to materials and methods under chemicals and reagents subtitle. Also delete what mentioned in the last sentence of introduction.

Response: The structural formula of the two insecticides have been transferred to “2. Materials and Methods” under “2.1. Chemicals and Reagents” subtitle, and showed in lines 93-94 and 106-107.

- Under 2.3. sample preparation: Please provide the method of extraction and clean up used for preparing the samples for residue analysis.

Response: This is a very good suggestion. We have added the methods of extraction and purification of thiamethoxam and fenvalerate and showed in lines 129-131, namely “Thiamethoxam and fenvalerate in cabbage, spinach and lettuce were extracted and purified according to Chinese National Standard GB/T 20769-2008 and Chinese Agri-culture Industry Standard NY/T 761-2008.”

- Please mention the name of analytical device and its specifications under point 2.5. Thiamethoxam and Fenvalerate Residue Determination.

Response: Analytical instruments have been added to 2.5 section, which was showed in lines 154-159, namely “The thiamethoxam contents in cabbage, spinach and lettuce were determined using an ACQUITY Ultra Performance liquid chromatography (Waters Corporation, USA) interfaced to a TSQ Quantum Ultra mass spectrometer (Thermo Fisher Scientific Inc., USA) according to Chinese National Standard GB/T 20769-2008. The fenvalerate contents were determined using an Agilent 6890N gas chromatograph (Agilent Technology Co. LTD, USA) according to Chinese Agriculture Industry Standard NY/T 761-2008.”.

- Conclusion is very long. Please make it short and concise to give a very specific information about major findings and their effects.

Response: Thank you for the comments. The conclusions have been revised and showed in lines 599-623.

- Also please revise the whole manuscript for some grammatical and language errors.

Response: Thank you for the comments. We have revised the grammatical and language errors of the manuscript and the changes are marked in red.

Reviewer 2 Report

The authos evaluated the Effects of Thiamethoxam and Fenvalerate Residue Levels on 2 Light Stable Isotopes of Leafy Vegetables.

The paper is well organized.

Some minor rmarks are follow.

Line 71.NH3...Please write in correct form

Line 176. g/m2...Please write in correct form

Your conclusions are more suited to the discussion, and you could include them in the discussion section. Please rewrite the conclusions focusing on the main findings in one paragraph.

Author Response

Reviewer 2

Comments and Suggestions for Authors

The authors evaluated the Effects of Thiamethoxam and Fenvalerate Residue Levels on Light Stable Isotopes of Leafy Vegetables.

The paper is well organized.

Some minor rmarks are follow.

Line 71.NH3...Please write in correct form

Response: Thank you for the comments. We have revised the incorrect form and showed in line 72.

Line 176. g/m2...Please write in correct form

Response: We have revised the incorrect form and showed in line 184.

Your conclusions are more suited to the discussion, and you could include them in the discussion section. Please rewrite the conclusions focusing on the main findings in one paragraph.

Response: This is a very good suggestion. The conclusions have been revised and showed in lines 599-623.